# Towards a One Health Food Safety Strategy for Palestine: A Mixed-Method Study

**DOI:** 10.3390/antibiotics11101359

**Published:** 2022-10-05

**Authors:** Said Abukhattab, Miriam Kull, Niveen M. E. Abu-Rmeileh, Guéladio Cissé, Lisa Crump, Jan Hattendorf, Jakob Zinsstag

**Affiliations:** 1Swiss Tropical and Public Health Institute, Kreuzstrasse 2, CH-4123 Allschwil, Switzerland; 2Faculty of Medicine, University of Basel, Petersplatz 1, CH-4001 Basel, Switzerland; 3Faculty of Science, University of Basel, Petersplatz 1, CH-4001 Basel, Switzerland; 4Institute of Community and Public Health, Birzeit University, Birzeit P.O. Box 14, Palestine

**Keywords:** Palestine: One Heath, food safety: antimicrobial resistance, zoonosis: foodborne pathogens, mixed methods

## Abstract

**Introduction:** Foodborne diseases, together with increasing antimicrobial resistance (AMR), pose a threat to public health in an era of huge challenges with climate change and the risks of zoonotic epidemics. A One Health approach to foster food safety is a key for improvement, particularly in complex socio-ecological systems such as in Palestine, to examine human–animal-environment interfaces and promote intersectoral action. **Objectives:** This study aimed to assess food safety from farm to public health toward an operational One Health strategy for Palestine. This study evaluates the food production (broiler production) and monitoring system to better understanding the zoonotic foodborne illnesses transmission and their resistance to antimicrobials. **Methods:** The transdisciplinary approach included multi-stakeholder discussion groups and field visits to broiler farms, slaughterhouses, and meat stores in the Ramallah and Al-Bireh and Jerusalem districts using a semi-structured observational tool. A survey with 337 poultry producers and workers in slaughterhouses and meat stores was conducted to assess hygiene knowledge, attitudes, and practices during broiler meat production. **Results:** The stakeholders point out various challenges along the food production chain in Palestine, such as a striking scarcity of public slaughterhouses, insufficient coordination between authorities, a gap between public and private sectors, and inconsistent application of the law. From observations, it appears that, unlike traditional broiler production, the public slaughterhouses and meat markets have effective hygiene, while large-scale farms implement biosecurity measures. Overall, surveyed participants reported that they are aware of zoonotic disease transmission routes and value hygiene standards. Semi-structured observations and survey results are contradictory. Observations indicate poor hygiene practices; however, the vast majority of broiler meat production chain workers claim that hygiene standards are met. **Discussion and Conclusions****:** Our study found that the overuse of antimicrobials, system fragmentation, insufficient infrastructure, a lack of regulations and controls, and poor hygiene practices are among the main obstacles to improving food safety in Palestine. Considering the risk of an important human health burden of food-related illnesses, enhancing food safety in Palestine is required using an integrated One Health approach. It is crucial to develop an integrated quality control system for food production along with promoting on-farm biosecurity and antimicrobial stewardship. Infrastructure, especially slaughterhouses and laboratories, must be built, training and education provided, and consumer awareness raised. As an important added value within a One Health strategy for better food safety in Palestine, research should be reinforced and accompany any future development of the food production monitoring system.

## 1. Introduction

The burden of foodborne disease hinders the achievement of the Sustainable Development Goals (SDGs), particularly the aspirations to end poverty, achieve food security, and ensure health and well-being [1]. Nearly one in ten people, about 600 million worldwide, becomes sick after consuming food contaminated with harmful bacteria, viruses, parasites, prions, or chemicals annually [1,2]. In addition to threatening public health, foodborne illnesses also impose a financial burden that hinders socioeconomic development by depleting healthcare systems and harming national economies, tourism, and trade [1].

Zoonotic pathogens can be transmitted to humans at all stages of the food production chain. For instance, *Salmonella* and *Campylobacter* , the most common foodborne bacteria, can be transmitted during food production, distribution, sale, or preparation if basic hygiene practices are not followed [3,4,5]. However, the main transmission route is generally considered to be foodborne via consumption of contaminated food [4,5]. Although improper preparation or handling at home, in food establishments, or markets account for a large proportion of food contamination, the World Health Organization (WHO) still assigns primary responsibility for preventing food contamination to food producers [3]. Nevertheless, contamination occurs along the entire food production chain, and disease transmission is sensitive to the growing and emerging threats from climate change and risks of zoonotic epidemics outbreaks [6]. The so-called “farm-to-fork” approach integrates all the processes from food production, transformation, transport, to marketing to assure a healthy product for the consumer [7]. A One Health approach extends the farm-to-fork concept to explicitly include human health and allows for assessing the full societal benefits of a systemic study of food-related health issues from a closer cooperation of human and animal health and other related sectors [8].

A One Health and transdisciplinary perspective, considering human–animal–environmental interfaces, is central in the context of food safety as it embeds food production systems in their environmental, socio-economic, and public health contexts. Integrated One Health approaches add value in terms of optimized human, animal, and environmental health compared to frameworks that focus on individual elements of food safety only [9]. WHO reports that prevention strategies implemented to reduce the prevalence of *Campylobacter* in live poultry have been associated with a similar decline in human cases [5]. This observation illustrates the interplay of the complex socio-ecological system and underlines the value of inter-sectoral action. To foster collaboration between different disciplines and also between science and society, transdisciplinary approaches promote dialogue between academic and non-academic actors [10]. Academic knowledge is complemented by valuing practical knowledge from the “life-world” to co-produce novel transformational knowledge [11]. Transdisciplinary approaches bring together academic and non-academic stakeholders. In this way, practical knowledge and perspectives of social domains complement scientific knowledge and contribute significantly to the success, quality, acceptance, and sustainability of interventions [11,12].

Despite the fact that foodborne illnesses in Palestinians have been rarely reported, the population is exposed to pathogens on a regular basis. In the Gaza Strip, *Campylobacter* and *Salmonella* were detected in approximately 5% and 2% of stool samples from children with acute diarrhea, respectively [13,14,15]. These *Salmonella* isolates were resistant to several antimicrobials, including 62% to 78% ampicillin resistance and 89% doxycycline resistance. [13,14,15]. A study by Adwan and colleagues reported a prevalence of *Salmonella*, *Escherichia coli*, and *S. aureus* of 25%, 95%, and 30%, respectively, in different types of meat in Palestine [16]. The reported resistance rates in Palestine were 59% and 80% for tetracycline, 59% and 51% for ampicillin, and 59% and 45% for nalidixic acid in humans and poultry non-typhoidal *Salmonella* (NTS) isolates, respectively. The authors conclude that the high prevalence of resistant NTS, especially the frequent resistance to fluoroquinolones, has become a serious problem in Palestine [17]. Moreover, the presence of antibiotic residues in up to 22% of raw dairy milk that reaches Palestinian consumers and in 24% of chicken breast samples collected from slaughterhouses in the Gaza Strip was confirmed [18,19].

The lacking capacity for zoonotic disease surveillance, the insufficient fight against antimicrobial resistance (AMR), and the low number of education and awareness events in the Middle East and North Africa call for a One Health initiative in these regions [20,21]. In the absence of relevant data on the risk of AMR from environmental sources, the Eastern Mediterranean and African regions are counted among the least contributors to global research findings on AMR in the environment [22]. To the best of our knowledge, food safety in Palestine has not been considered from a One Health perspective [21]. This research aims to identify the structure, strengths, obstacles, and limitations of the food production monitoring system in Palestine through multi-stakeholder discussion groups. Complementing the qualitative study, we evaluated hygiene practices at broiler farms, slaughterhouses, and meat stores in Palestine through quantitative and semi-structured observational studies. This assessment forms the basis for specifically adapted future developments of the monitoring system and associated improvements in food safety in Palestine using the One Health approach.

Conflict zones, such as the ongoing conflict situation in Palestine, can have a negative impact on social structures, including the family and community. Such conflict scenarios can be extremely complicated and have an impact on aspects such as the environment, nutrition, economy, and psychosocial factors [23]. The cost of war and violence results in environmental concerns in addition to psychological misery. Efforts to improve health care and public health will be hampered by this complicated socio-ecological system, which will also hinder social performance and economic progress [23,24]. In Palestine, 29.2% of the population, or more than one in three, were deemed to be living in poverty in 2017. In 2017, the poverty rate in the Gaza Strip was 53%, which was more than four times higher than the rate in the West Bank [25]. By 2020, the agricultural sector’s value added in Palestine reached 7.1% of the gross domestic product [26]. Broiler production was chosen as a prime example of food production in Palestine. In 2019, Palestine produced more than 70 million broiler chicks [27,28]. In 2020, poultry meat production accounted for more than 60% of the country’s total meat production at 42,728 tons [29]. A better understanding of the structure, strengths, obstacles, and limitations of the food production monitoring system in Palestine and the context in which it is embedded is essential for developing the monitoring system and related improvements to food safety in Palestine.

## 2. Results

### 2.1. Stakeholder Points of View on the Food Production Monitoring System

Various interest groups share numerous challenges and issues along the food production chain in Palestine. Some of the most commonly cited concerns include inadequate hygiene practices, scarcity of veterinary services, laboratories and public slaughterhouses, the uncontrolled use of antimicrobials, insufficient monitoring measures, coordination between authorities, communication between public and private sectors, and inconsistent application of the law. While stakeholders agree on some issues, such as the urgency of containing the spread of AMR, the multi-stakeholder discussion groups also revealed some major differences in perceptions of the various interest groups mainly related to lab testing and services.

The multi-stakeholder discussion group outcomes for each of the five key themes (“The current system for monitoring food production”, “Regulatory authorities with responsibilities relevant to food safety”, “Public health”, “Available infrastructure and capacity building”, and “Political and legal context”) are presented in detail below (Appendix A).

#### 2.1.1. The Current System for Monitoring Food Production

The multi-stakeholder discussion groups reveal that certain surveillance tools, such as monitoring proper food transport or testing for pathogens, already exist but with some limitations. For instance, the Ministry of Health asserts that the proper transportation of food is monitored: *“There was a case of transporting milk in plastic tanks that did not meet the conditions and standards, so a decision was given to destroy the milk”*. Meanwhile, one of the participating poultry farmers raises the issue of improper meat transportation: “*The meat transporting is not done in the right and proper way. Sometimes it is done in private cars or not a refrigerated car*”.

Regarding examination for pathogens, the discussion groups indicate that pathogen testing is practiced in suspected cases by authority bodies. *“In the central veterinary laboratory, our main function is diagnostic services through surveys of certain animal diseases or animal products. Any veterinarian who suspects important diseases sends us samples for examination”*. Challenges such as disease outbreaks seem to overwhelm the monitoring system in Palestine. Some stakeholders point out that the surveillance system is not able to contain disease outbreaks because identified disease cases are often not reported, properly identified, or followed up. According to a private veterinarian, follow-up of disease cases is not done, resulting in a *“permanent disorder”*. One of the participating private veterinarians agreed, describing the monitoring system fundamentally as *“weak”* and its outcome as *“almost zero”.*

Stakeholders particularly criticize the poor implementation of monitoring tools such as the licencing of farms or veterinary health certificates. The issue of licensing, especially of small informal farms, so-called random farms, was raised by several stakeholders, including representatives of poultry breeders and the Ministry of Health: *“Random farms mean few are licensed, and therefore, it is difficult to control the market”*. The Ministry of Health explained that there were random farms that neither the veterinary services nor the health departments knew about and stressed the importance of monitoring the issue of licensing.

#### 2.1.2. Regulatory Authorities with Responsibilities Relevant to Food Safety

Stakeholders from the academic, public, and private sectors agreed that some tasks are clearly assigned to the respective ministries. One participant asserted that there was no interference between the powers of the regulatory authorities. The role distribution between authorities, as confirmed by various stakeholders, assigned responsibility for live animals to the Ministry of Agriculture, which is in charge of field controls. The Ministry of Local Governance was responsible for slaughterhouses and once slaughtered, further processing up to the final product was the responsibility of the Ministry of Health. At the same time, the Ministry of National Economy supervised the marketing of food products.

While some responsibilities are assigned to a single regulatory authority, others were shared by multiple authorities. For instance, one public sector representative stated that although the Ministry of Health is the regulatory authority for food safety, it worked closely with other ministries for this purpose.

Other shared responsibilities include import and export or the health sector, which is in charge of three ministries, Local Governance, Health, and Agriculture. One stakeholder was struck by deficient coherence between these ministries to jointly decide and implement measures. In contrast, a representative of the Ministry of Health highlighted the presence of a joint committee between the Ministry of Agriculture and the Ministry of Health called the *“Zoonotic Committee”* as well as a monthly epidemiological report in which both ministries participated. *“Recently, there was a Brucella outbreak in Ramallah area which extended to one of the refugee camps. Direct coordination was made between the United Nations Relief and Works Agency for Palestine Refugees (UNRWA), the Ramallah Health Directorate, Preventive Medicine Department and Environmental Health Department at Ministry of Health and Veterinary Service Directorate at Ministry of Agriculture to reach the source of infection”.*

Several stakeholders observed not only overlap but also conflict among regulatory authorities with adverse consequences, such as delayed application of the Palestinian food strategy which was issued in 2017 and AMR national action plan that was issued in January 2020. Both of them are still in infancy and need more participation, collaboration, understanding, and support at the national level.

#### 2.1.3. Public Health

Stakeholders advocated for worker and consumer health and the need for prevention, surveillance, and control of foodborne pathogens. Some interest groups pointed out that not only consumers but especially workers who are in direct contact with animals and veterinary medicines are exposed to health risks: *“**Hormones are used to gain weight in the fattening farms, which affects human fertility**”*. Poor hygiene or contamination by water was, for example, a possible cause of Salmonella in poultry, explained the Ministry of Health. However, as multiple stakeholders agree, the adequate preparation of animal food products also plays a crucial role in the prevention of zoonotic disease transmission; the participants agreed the best example of this issue in Palestine is Brucellosis *“**We are supposed to convince the consumer to eat what is pasteurized. If it is implemented, 80% of brucellosis cases in Palestine will be ended**”.*

In addition to foodborne pathogens, stakeholders were also concerned about antimicrobial residues in foods marketed for human consumption. As the AMR national action plan committee warns, humans may develop resistance to antimicrobials due to antibiotic residues in animal meat. One participant commented on this issue by saying: *“**I suspect a high percentage of products of animal origin, whether poultry, sheep or cattle, contain antibiotic residues**”*. Antimicrobials can directly impact human health, as various interest groups agree, but the most feared impact is AMR, the emergence of bacteria that are resistant to antimicrobial drugs. According to the Ministry of Health, the issue of antimicrobial resistance has not yet been studied at the Palestinian level: *“In Palestine, we need to strengthen AMR surveillance by establishing AMR national surveillance system”*.

While AMR occurs naturally over time, as the public health laboratory acknowledged, the misuse and overuse of antimicrobial drugs accelerate its emergence. The AMR national action plan committee was in agreement with this, stating that “*antibiotic misuse leads to resistance*” and emphasizing that the prescription and use of drugs in veterinary and human medicine were critical. The Ministry of Health explained the misuse and overuse of antimicrobials through poor prescribing practices, improper drug selection, incorrect dosing, and self-medication practices: *“The person (health professionals) must have adequate attitudes and awareness; for example, use of third-generation antibiotics in the onset of disease without the use of sensitive test results”*. In animal production, as the as the Ministries of Agriculture and Health pointed out, misuse and overuse of antimicrobial medicines occur especially when they are administered for non-therapeutic purposes, such as prophylaxis or animal growth promotion: *“There are no restrictions on dispensing antibiotics or growth promoters; they are often handed out without a veterinarian’s prescription”.*

One of the participants from the government sector stated, “*In 2016, 201 samples from farms of laying and broiler chickens came to us on the basis that the farms had high mortality and no antibiotics were effective. When a sensitivity test was conducted, we found that more than 50% of isolated E. coli were extremely resistant to all antibiotics”*.

To slow the spread of AMR, stakeholders requested measures including: (1) Good manufacturing practices for antimicrobials; (2) avoidance of antimicrobial use in animal farms unless for treatment purposes; (3) prescription of antimicrobials for humans exclusively by physicians; (4) improvement of the antimicrobial surveillance system in Palestine; (5) networking of laboratories at the national level; (6) inspecting, testing, and controlling antimicrobial residues in animal feed products and food sources; (7) development of a national awareness program. In addition, stakeholders questioned whether enough culture and susceptibility testing was being done and advised for the wider use of these tests. The department of epidemiology and preventive medicine acknowledged that culture-susceptibility testing was difficult to perform for each individual case in human medicine and that the procedure was performed in hospitals but not in private clinics.

#### 2.1.4. Available Infrastructure and Capacity Building

Farmers and veterinarians did not seem to have much confidence in the sample collection, handling, and testing procedures followed in animal health laboratories and called for capacity building. Veterinarians were even convinced that samples did not arrive at the laboratory, and if they did, they were neglected and not examined: *“I sent 21 chicks to the government laboratory to check for mycoplasma after a week, and the answer was (the sample is not enough)”.* Another veterinarian said, *“I have not sent samples to the laboratory for more than 3 years... Samples were sent, and the result was that pathogens were not isolated, and all results were negative despite the high mortality rate with clinical symptoms, and in the end, all animals were dead. They have laboratory equipment, but they don’t use them”. “The vet himself does not trust in the government laboratory”.* The Ministry of Health agreed partially with the veterinarians, and called for a general expansion of diagnostic capacity in laboratories. This was supported by other stakeholders, including the public health laboratory. The public health laboratories complained about inadequately trained personnel, lack of certain equipment and materials, and insufficient national coordination, which entails duplicate sampling. The Palestinian authorities should better coordinate collaboration between the veterinary laboratory, public health laboratories, and laboratories at universities.

The discussion groups agreed that *“more than 70% of slaughtering takes place outside public slaughterhouses”*, which is explained by the lack of official slaughterhouses and limited control of the existing ones since not all cities, villages, and refugee camps in the country have poultry slaughterhouses. The Ministry of Agriculture as well as the Ministry of Health condemn slaughtering outside public slaughterhouses, arguing that hygiene practices in traditional slaughterhouses, known as *“Natafats”*, are not controlled. The ministries share concerns about contamination due to poor hygiene practices in traditional abattoirs. *“We must improve some practices during the slaughtering process, such as reusing materials and tools and personal hygiene in the Natafats, which will reduce carcass contamination*”. While the ministries require the presence of a veterinarian at the slaughterhouse to grant permission for operation, even in public slaughterhouses, stakeholders criticize the absence of veterinarians during the operations of these slaughterhouses: “*To improve control of the slaughter process, slaughterhouses should be run privately but remain under government control*”. A representative of the Ministry of Health suggested, “S*laughterhouses should be equipped with cameras and be linked to the Ministry of Agriculture. In this way, not only could the veterinarians’ presence be monitored, but also compliance with their duties*”.

#### 2.1.5. Political and Legal Context

From the discussion groups, it appeared that one obstacle to food safety in Palestine was the gap between the private and government sectors. Stakeholders argued, for instance, that food safety strategies failed in the past because only the government sector was involved. One of the poultry breeders, who has worked in poultry production for five years, raised the specific issue of missing communication between farmers and the Ministry of Agriculture. This poultry breeder criticized that “*the Ministry of Agriculture does not support farmers in matters of health, product safety, and poultry farms protection and is therefore of no concern to farmers, it is even non-existent*”.

“*If there was an insurance and compensation system for farmers, a control system would be superfluous*”, argues one of the poultry producers. Another farmer adds, “*Sometimes we look for medications that are the cheapest for treatment. Some medications have better treatment effectiveness, but if I use them, I lose the profit of my work.” “Indeed, the lack of compensation forces farmers to slaughter animals regardless of the health status of the flock*”, one of the veterinarians justified. Veterinarians should also be subject to legal protection, according to one of them: “*My colleague works in a slaughterhouse and after eliminating a calf unfit for human consumption; he was attacked by having his car set on fire and shot at. But currently, there is no criminal law that would deter criminals from committing such acts*”. Another participant representing the government sector said, “*We frequently find antibiotics residues in broiler meat samples collected from traditional slaughterhouses. However, we are unable to proceed because we lack legal provisions to take any enforcement action against these facilities*”.

Obstacles to the implementation of Palestinian law and deficient import controls at border crossings were cited as another impediment to food safety in Palestine. “*The Palestinian law, which was built, based on the Jordanian law, has been amended and updated many times, and this amendment has led to the weakening of the Palestinian law, as stated by one of the participants*”. The problem of smuggling, especially the large number of uncontrolled poultry imported from Israel to Palestine, was raised by various stakeholders, such as the Ministry of Health and representatives of poultry production. The Ministry of Health referred to the regional differences in the implementation of the law in Palestine: *“The problem of smuggling is due to weak law enforcement outside the cities or on the border areas, which constitute more than 60% of the Palestinian Authority’s land”*. In all meetings, there was consensus among stakeholders that the law should be applied to all Palestinian Authority territories with the same effectiveness.

### 2.2. Semi-Structured Observational Study: Hygiene Practices along the Broiler Production Chain

#### 2.2.1. Hygiene Practices in Broiler Farms

Large-scale farms visited had biosecurity and modern technologies at their disposal (Appendix A). On the other hand, farms with smaller production lacked automatic ventilation systems, temperature control, or even closed buildings (Appendix A). While floors were usually concrete, sometimes they were covered with loose, dry bedding, soil, or both. Nearly all farmers reported that dead chickens were either fed to pets or deposited in municipal open dumps (Appendix A), and that manure was used as fertilizer for olive trees (Appendix A). Farms were run by workers who often live on site. The bedroom, kitchen, and toilet were usually located near the chicken farm, often even in the same building, separated only by a door. On only one large- and one small-scale farm visited were the workers living in separate buildings. All but one of the facilities had running water, but soap was rarely seen, and a towel was available in only one establishment. Kitchen and toilet cleanliness at the time of visit ranged from dirty to very clean and smoking shisha or cigarettes inside the rooms was the norm (Appendix A) and (Appendix A). Overall, workers’ personal hygiene and accommodation cleanliness seemed better in larger agencies than in small farms.

#### 2.2.2. Hygiene Practices in Poultry Slaughterhouses

The procedure of slaughtering was similar in traditional abattoirs and public slaughterhouses, but public slaughterhouses made use of modern technologies at all processing steps, while traditional abattoirs relied on simple equipment only. Slaughter in traditional slaughterhouses depended directly on demand as consumers came to the abattoir to select the live animal that is kept right inside the abattoir. Public slaughterhouses started work at 10:00 p.m., working all night so that the fresh poultry could be transported to the markets before they opened around 6:00 a.m. The public slaughterhouse we visited had the capacity to process up to 30,000 broilers every night (Appendix A).

#### 2.2.3. Hygiene Practices in Poultry Meat Stores

Selling points of poultry meat in Ramallah city and surrounding villages were highly similar. Chicken was kept fresh by cooling but was never frozen, as there was no demand for frozen poultry meat. Poultry was sold either as whole or cut into breasts, thighs, wings, kidneys, and livers (Appendix A). Overall, the poultry meat stores visited appeared clean and well maintained, and hygiene, such as hand washing, seemed adequate.

### 2.3. Knowledge, Attitudes, and Practices (KAP) of Hygiene among Broiler Production Chain Workers

#### 2.3.1. Sociodemographic Characteristics of the Study Population

Overall, 337 workers in the broiler meat production chain from 175 locations, from farms to meat stores, participated in the survey. The sociodemographic characteristics of the poultry meat chain interviewed are summarized in Table 1. It is noticeable that almost all respondents were male (98.2%) and lived in the countryside (89.3%). Most were married (69.9%) with up to 13 family members. The majority had at least completed junior high school (94.3%), and 27.0% had attended university. Respondent experience in broiler production differed greatly, varying between 6 months and 60 years. The monthly income median was 4000 NIS (New Israeli Shekel), equivalent to about 1200 U.S. dollars, and varied strikingly, with minimum of 1000 NIS and maximum of 8000 NIS.

#### 2.3.2. Reported KAP of Hygiene in Broiler Production

Interview results suggested that respondents were aware of zoonotic disease transmission routes and valued hygiene standards such as thorough hand washing or personal protection, especially when in contact with sick or dead chickens (Table 2). The survey results also revealed that stricter hygiene measures and regular disinfection of workplaces were introduced since the SARS-CoV-2 pandemic (Table 3). With 84.3% of respondents having heard of it before, zoonotic diseases seem to be a concept for most participants, while 6.8% of them were infected with one of the zoonotic diseases related to their works. About 94% of respondents believed in promoting proper hand hygiene and 82.5% considered that personal protective measures when in contact with poultry could prevent zoonotic disease infection. According to the surveys, about 11.3% washed their hands without soap, while the rest used soap and or disinfectants. In fact, 95% declared that they had introduced frequent hand washing since the emergence of the SARS-CoV-2 pandemic. In addition, 50.5%, 65.0%, and 68.0% disclosed that they had adopted wearing overalls, masks, and gloves, respectively, since SARS-CoV-2 emerged. In addition, 75.8% reported using a tissue when sneezing since the outbreak of the pandemic (Table 3).

While 264 of 337 respondents believed touching sick or dead poultry could transmit a zoonotic disease, 164 admitted to touching sick or dead poultry with their hands in the past month. Of these 164, 88 respondents reported taking preventive measures, such as thorough hand washing after contact with the animal. A total of 95.0% of respondents believed that increasing the frequency of cleaning and disinfection and ensuring adequate ventilation in common, spaces could prevent zoonoses. Since the emergence of SARS-CoV-2, there has been a 79% rise in sterilization with disinfectant after finishing work, an 82.1% increase in disinfection frequency, and an 85% increase in ventilation in the workplace (Table 3).

About 22.9% of dead chickens and slaughter waste were fed to pet animals, without any treatment, while 22.9% were released in the wild, and municipalities collected 32.3%. The rest reported that they burn or bury dead animals. Few respondents specified that they burned dead chickens if they had a disease and fed them to pets if they were healthy before they died.

An association between poultry farmers’ practices and their attitudes could not be demonstrated in the survey results. Table 4 shows the frequencies of respondents by attitude and related practice regarding hand hygiene and contact with poultry. These results indicated no strong association between attitudes and related practices. Assuming that respondents’ practices correlate with their attitudes, we would expect consistent responses to the questions “Do you think promoting proper hand hygiene can prevent infection with zoonotic diseases?” and “How do you wash your hands?”, for instance. In this case, we would expect that respondents who believe that proper hand hygiene prevents zoonotic disease transmission are more likely to wash their hands with soap and or disinfectants than respondents who do not believe that proper hand hygiene does prevent zoonotic diseases. A univariable logistic regression estimated a non-significant OR of 0.82 (95% CI [0.18, 3.66]) was associated with the responses to the two questions, and hand hygiene practices appear to be independent of farmers’ attitudes toward hand hygiene. The comparison between individuals who believe that touching sick or dead poultry can cause zoonotic infection and those who do not believe or do not know reveals a non-significant OR of 0.69 (95% CI [0.41, 1.17]) related to the response to the question “Did you touch sick or dead poultry with your hands in the last month?”. With an OR of 1.15 (95% CI [0.5, 2.6]), no association can be found between the questions “Do you think personal protective measures when in contact with poultry can prevent infection with zoonotic diseases?” and “Did you take preventive measures when you touched sick or dead poultry?”. Based on the questions analyzed in this study, we could not validate the consistency between respondent attitudes and practices.

#### 2.3.3. Determinants of Broiler Production Chain Workers Knowledge about Zoonotic Diseases

Our analysis identifies education level and years of experience working in the broiler meat production chain as likely determinants of poultry farmer and meat worker knowledge about zoonotic diseases. The examination of the proportion of respondents who have heard or have never heard of zoonotic diseases by education level suggests a relationship between respondent education and their knowledge (Figure 1). For instance, 3 out of 5 illiterate participants (60.0%) had not heard of zoonoses, while the majority of individuals within each other educational category knew about zoonotic diseases. About 71%, 66%, 90%, and 96% of respondents who graduated from elementary school, junior high school, senior high school, and university, respectively, had heard of zoonotic diseases. For participants with more than nine years of education (completed senior high school or university), zoonotic diseases seemed to be a familiar concept. Years of experience working in broiler meat production appeared to be another determinant of zoonotic disease knowledge.

As shown in Figure 2, workers (farmers and meat workers) who knew about zoonotic diseases tended to be more experienced in broiler meat production (median years of experience working in the broiler meat production chain = 9 years) than those who had never heard of zoonotic diseases (median years of experience = 4 years). The results of a multiple logistic regression supported the suggested dependence of zoonotic disease knowledge on education level and years of experience working in the broiler meat production chain. The odds of having heard of zoonotic diseases were 4.54 (95% CI [2.28, 8.99]) and 10.88 (95% CI [3.69, 32.06]) times higher for education levels of senior high school and university, respectively, compared to individuals who were illiterate or whose highest level of education was elementary or junior high school. Further, for every one-year increase in experience in working in broiler meat production, the odds of having heard of zoonotic diseases increased by a factor of 1.13 (95% CI [1.06, 1.20]). Thus, our model suggested that education level and years of experience working in broiler meat production were likely determinants of zoonotic disease knowledge.

## 3. Discussion

### 3.1. Toward an Operational One Health Strategy for Palestine

Although a national food safety strategy was developed, unclear roles and poor communication among regulatory authorities as well as between private and public sectors appear to be impeding progress in improving food safety in Palestine. The division of responsibilities among ministries, as determined by the national committee for the formulation of the Palestinian strategy for food safety, is consistent with the reports of the multi-stakeholder discussion groups we conducted [30]. Our results regarding the role distribution between regulatory authorities could thus be confirmed. The Palestinian food safety strategy agrees that the main responsibility for monitoring food safety, including hygiene guidelines for service providers, food safety inspections, and prohibiting the marketing of unregistered foods until they have obtained the necessary permits, or disease control measures, lies with the Ministry of Health [30]. However, as mentioned in the discussion groups, the Ministry of Health is expected to collaborate with other stakeholders to develop control mechanisms, ensure workplace safety, or inspect imported food [30]. At the same time, the responsibility to regulate imports and exports of foods is assigned to the Ministry of Agriculture [30]. We assume that these shared responsibilities bear the risk of misunderstandings and often fail due to insufficient communication and coordination between regulatory authorities and poor communication between the private and government sectors.

Our findings suggest that there is a lack of adequate resources and personnel to implement a One Health strategy for improved food safety in Palestine. These findings are consistent with the results of Abuzerr and colleagues and Al-Khatib and colleagues who identified a lack of policy coherence, poor governance and leadership, limited financial resources and the lack of explicit national food safety requirements and standards as major barriers to implementing the One Health integrated surveillance system [31,32]. In the specific context of implementing *Brucella melitensis* control programs in Palestine, for example, based on the FAO report, the program is challenged by weak infrastructure, scarce resources, insecurity, political instability, vaccine quality, farmer attitudes, and deficient traceability of animal movements [33]. Under these circumstances, the organization notes, improving occupational and food hygiene, monitoring disease outbreaks in humans and animals, and implementing animal movement control are particularly challenging. A test and slaughter policy, for instance, should only be considered if farmers are fully compensated for their slaughtered animals, cooperate, and accept the slaughter policy [33].

Rigorous, evidence-based strategies should guide hygiene practices in Palestinian animal food production from farm to final food preparation. It has been shown that on-farm biosecurity measures could reduce the prevalence of *Campylobacter* infection in broilers by more than 50% [34,35]. Thus, renovating traditional poultry farms in Palestine or constructing new buildings should be supported and encouraged. A simple measure such as the use of screens to keep flies out has been associated with a reduced prevalence of *Campylobacter* from 51.4% in the control houses to 15.4% in the intervention houses [36]. We hypothesize that such a reduction in infection of zoonosis pathogens in live poultry will also reduce the risk of transmission to humans by reducing occupational hazards for workers who are in direct contact with animals or meat products. On the other hand, better product quality prevents the transmission of zoonoses to consumers.

The stakeholders expressed particular concern about misuse and overuse of antimicrobials and the associated acceleration of antimicrobial resistance. The researchers agree that critical steps for mitigating antimicrobial resistance and maintaining future efficacy of antimicrobials requires a comprehensive, system-wide, multidisciplinary, and multisectoral strategy to promote, monitor, and evaluate judicious use of antimicrobials [22,37]. Appropriate precautions, such as biosecurity measures and a national action plan for antimicrobial stewardship (AMS), may mitigate the impact of foodborne pathogens and AMR, but these challenges remain and require an integrated One Health approach. Biosecurity measures reduce but cannot exclude the risk of *Campylobacter* infections in broiler flocks because chickens are exposed to “constant contamination pressure” [38]. Van de Giessen and colleagues record that positive cases reappeared after the successful elimination of *Campylobacter* [35]. When it comes to AMS, the United Nations Interagency Coordination Group on Antimicrobial Resistance (IACG) cautions that most countries do not fail to develop a national action plan for AMS but rather to implement and sustain it [39]. Challenges to implementing AMS include technical capacity, necessary finances, political will, and regional cooperation to prevent poor implementation of measures in one region from undoing progress in other regions [39]. In the long term, IACG insists governments must allocate resources to implement their national action plans to ensure sustainability and achieve effective AMS [39]. Meanwhile, IACG asserts that allocating resources to address AMR is one of the most profitable investments a country can make [39].

During our research, we realized that incorporating legal, political, and social dimensions as part of a One Health initiative for improved food safety in Palestine is indispensable. Current legislation regulating food safety in Palestine is fragmented and does not cover all aspects of food safety at all levels of the food chain [30]. Due to the Israeli occupation, uncontrolled trade takes place on borders and in “Zone C”, which covers more than 60% of the West Bank where Palestinian authorities are prevented from enforcing their policies [30]. Given decades of conflict, war, and unrest, developing the political will for international cooperation for disease prevention and AMS presents one of the greatest challenges in Palestine.

A limitation of this study is that the composition and participation of the multi-stakeholder discussion groups have an impact on the outcomes. We tried to be inclusive by inviting all relevant sectors. However, the multi-stakeholder discussion groups had an imbalance of power since not all invited interest groups were equally active and engaged. For instance, in the Ministry of Local Governance, the municipalities did not actively participate in the first discussion group, leaving the discussion early.

### 3.2. Methods to Assess Hygiene Practices

Focusing on farms and broiler production, we propose utilizing various approaches to evaluate hygiene practices. Even though some direct observations were confirmed by the questionnaire responses, we noted poor agreement of survey results and data collected through observations. Observations and interviews agreed that dead chickens were most often fed to pets or disposed of at the municipal dump. In contrast, observations and survey results were contradictory on hand washing, protective clothing, disinfection, and ventilation at workplaces, and visitor hygiene regulations. According to the surveys, about 90% reported to wash their hands with soap, but we rarely found soap near the sink during site visits. A significant percentage of respondents reported wearing protective clothing, but workers were rarely wearing uniform clothing, tall boots, or masks during our visits.

Although field observations are considered one of the most appropriate tools for measuring hygiene behavior at the community level, their validity and reliability must be questioned [40]. In a few cases, our field visit was announced beforehand, so staff were present to show us around. This needs to be considered as the announcement could allow workers to prepare and adapt, for instance, getting rid of dead animals or cleaning the facilities. Observation at a single time point might miss individual behavior that varied over time [41]. However, because most of our observations examined facilities and equipment, such as the presence of a disinfecting footbath or the type of bedding in the poultry farm, they were not particularly susceptible to variability. In contrast, the reliability of observations related to the cleanliness of toilets, for example, maybe more vulnerable. Although time-consuming and costly, repeated observations could increase the reliability and validity of semi-structured observations, provided that farms became accustomed to the presence of researchers.

Over-reporting of good KAP may limit the validity of survey results on hygiene practices in broiler production. As reported in several studies assessing sanitary practices, we assume that KAP perceived as good tends to be over-reported in surveys [40,42,43]. The resulting overestimation of the frequency of good hygienic KAP affects the validity of survey results in that they may not accurately reflect reality. To avoid this point, a written questionnaire could also be considered instead of a verbal interview so that participants would not be influenced by direct interaction with interviewers. In addition to validity, the generalizability of surveys must also be questioned due to the restricted study area and the corresponding study population. The evaluation of associations between survey responses is limited by the sample size. For instance, the low number of respondents affected by a zoonotic disease transmitted by chickens, only 23 interviewees, makes it difficult to identify KAP as a risk factor. To examine KAP as risk factor for health outcomes, the study area and thus the sample size would need to be dramatically increased.

While the low correspondence between observations and survey results is noteworthy, we expected such inconsistencies due to our study design. We also assume that respondents may be over-reporting good hygienic KAP. Al-Khatib and Al-Mitwalli noticed in 2009 a similar divergence between survey results and field observations regarding food sanitation practices in restaurants of Ramallah and Al-Bireh district of Palestine [31]. For example, 76.5% of respondents reported washing their hands with soap, while the authors found that in 14.2% and 37.3% of cases, respectively, there was no cleaning material near the hand-washing sink in the kitchen and the toilet. Generally, data collected through direct observation of hygiene practices, as Curtis and colleagues in Burkina Faso, Manun’Ebo and colleagues in Congo, and Garayoa and colleagues in Spain, are less prone to overestimate the frequency of good practices than those through questionnaires [40,42,44]. Nevertheless, the authors emphasize that there is no gold standard for measuring hygiene practices, because just as participants can adjust their responses to a questionnaire to correspond to what they perceive as good, they can also adjust their behavior in the presence of observers to show an image they consider desirable.

### 3.3. Impacts of Food Supply Chain Actor KAP on Public Health

To the best of our knowledge, this is the first study attempting to describe the knowledge, attitude, and prevention practices of zoonotic disease in Palestine. Interest groups of the multi-stakeholder discussion groups emphasized that KAP are critical to preventing foodborne pathogens and fighting AMR at all levels of the food supply chain, from farm to fork. Farmers are considered end users of antimicrobials, which explains the importance of their KAP to mitigate the spread of AMR [45]. Similar to our results suggesting that farmer and meat worker education level and years of experience might be related to KAP of hygiene in broiler production, Hassan and colleagues demonstrate that in Bangladesh, farmer socioeconomic demographics, such as education, source of income, and age, have a major impact on KAP in the context of antimicrobial use and AMR [45]. Wambui and colleagues in Kenya and Martins and colleagues in Portugal also found a significant effect of education level on hygiene practice among meat workers [46,47]. Thus, their findings confirm our hypothesis that sociodemographic characteristics are likely determinants of KAP. The KAP results in this study revealed that personal hygiene, such as proper hand washing, is close to those reported by Osaili and colleagues in Jordan and higher than those reported in Kenya, Turkey, and Egypt [46,48,49]. Conversely, a higher proportion was reported by Neil and colleagues, who found that all South African meat workers who took part in their study indicated that they always washed their hands with soap while at work [50].

We also looked at knowledge of zoonoses diseases, and a high percent of respondents had heard of the term “zoonoses” prior to this study. These results are close to those reported in India (80%) and lower than those reported in Greece (99%) [51,52]. While studies in Malaysia and Nepal found that the proportions of prior knowledge of zoonotic diseases among the farmers were 42% and 74%, respectively [53,54].

The global economy and health have both been severely impacted by the SARS-CoV epidemic. Even though most firms were completely shut down due to government regulations, the food industry across the supply chain still operates to feed the nations. In such a challenging time, maintaining worker safety and health is essential, as is upholding a high standard of food safety and consumer confidence [55]. Personal protective equipment (PPEs), along with good personal hygiene and hand washing habits, are strongly recommended by health institutions such as WHO and the Centers for Disease Control for food business to minimize the spread of both cross-infection (SARS-CoV) and cross-contamination (food safety) [56,57]. Therefore, it is now more pressing than ever to have higher compliance with hygiene practices and protective measurements [55,57]. This strictness in applying hygiene practices in markets and meat factories, along with the hygiene awareness program to limit the spread of coronavirus within the Palestinian territories, was reflected in our KAP study that was carried out in the midst of the pandemic.

## 4. Methods

The study used methods to capture attitudes, views, opinions, interests, and needs of all relevant stakeholders involved in the food production chain. The exchanges between these stakeholders were facilitated, and their coordination and cooperation promoted. For these purposes, the study combined the use of multi-stakeholder discussion groups and semi-structured observations (Figure 3).

Furthermore, this study examined the hygiene practices of broiler meat production chain workers and the underlying knowledge, attitudes, and practices (KAP). An attempt was made to explain and understand current hygiene practices in broiler production. Of particular interest was the coherence between attitudes and practices in broiler production. In addition, sociodemographic characteristics, namely education level and years of experience in farming, were tested as determinants of the workers’ knowledge about zoonotic diseases. At best, broiler meat worker KAP can be identified as risk factors for health outcomes, measured by the proportion of respondents affected by a zoonotic disease transmitted by working in the broiler meat production chain (Figure 3).

### 4.1. Multi-Stakeholder Discussion Groups

This research projected adopts principles of transdisciplinarity to assess stakeholder points of view on the food production monitoring system in Palestine by incorporating contextual conditions, interests, knowledge, and expectations. The “multi-stakeholder discussion group” tool provided by the transdisciplinarity network (TD-net, transdisciplinarity.ch), a forum initiated by the Swiss Academies of Arts and Sciences, served as a guideline for the stakeholder meetings. This method brought together stakeholders and civil society to identify the problem at hand and facilitate a common understanding of different perspectives, mutual learning, successful co-production of knowledge, and building trust [58].

Discussion groups were conducted at Birzeit University in Palestine between July 2021 and February 2022 and included the public, private, and academia sectors. After identifying the social groups relevant to the project, representatives of key institutions, associations, networks, and stakeholders were invited to participate in a discussion group in July 2021. Representatives of the public and academic sectors attended the multi-stakeholder discussion group on 7 July 2021, composed of 11 stakeholders. Eight stakeholders involved in or overseeing food production were invited to participate in a second-day multi-stakeholder discussion group from 8 July 2021. These two initial discussion groups clarified the roles of the stakeholders and identified the various perspectives on the Palestinian food production monitoring system. The first multi-stakeholder discussion group included stakeholders from the public sector and academia to create the most favorable climate for authentic reporting. While participants in the second discussion group were private interest groups and stakeholders directly involved in monitoring food production. A third multi-stakeholder discussion group with 15 participants in February 2022 sought dialogue between the public and private sectors, academia, and civil society. Participant details in the three meetings are described in Appendix A.

The participant sociodemographic characteristics were considered. Participants ranged in age from 30 to 70 years, with between 5 to 45 years of experience in their field. The majority of participants in the first and second stakeholder discussion groups were male, while 30% of participants in the last discussion group were female.

### 4.2. Semi-Structured Observations

To assess hygiene practices, a semi-structured observational tool was based on a survey developed by Hawileh (2012) (Appendix A) [59]. This tool served as a basis for reporting hygiene practices at broiler farms, abattoirs, and meat selling points. Between 8 and 17 February 2021, a total of five farms, three abattoirs, and two meat stores within Ramallah area were visited. Sites were chosen so that the various types of farms and abattoirs, from traditional small-scale to modern, highly technological large-scale agencies were taken into account. Observations were not limited to hygiene practices during work, but also considered essential infrastructure: buildings and facilities, water availability, waste management, and worker personal hygiene. All field visits and observations were documented with handwritten notes and, when permission was provided, supplemented with photographs.

### 4.3. Surveys on Chicken Meat Production Chain Worker Hygiene Knowledge, Attitude, and Practices

A cross-sectional survey looked at the knowledge, attitudes, and practices of workers involved in the broiler meat production chain under the hypothesis of inadequate implementation of hygiene practices in broiler production in Palestine. Ramallah and Al-Bireh and Northwest Jerusalem districts were chosen as the study area so that a complete analysis of the broiler production chain, from breeding farms to slaughterhouses and meat stores, could be implemented. Ramallah and Al-Bireh governorate is the administrative and economic capital of the Palestinian Authority and is among the governorates with the most breeding broiler chickens, with nearly 10 million birds in 2021 [60].

Based on data from the Palestinian Ministry of Agriculture and Ministry of Local Government, the study population was designated as all broiler keepers, and slaughterhouse and meat store workers in Ramallah and Al-Bireh governorate, and Northwest Jerusalem region (175 sites in total). The study excluded farms that did not raise broiler chickens in the preceding year (May 2020 to May 2021). The structured questionnaire was based on a survey developed by Hawileh (2012) [59]. With a response rate of 85% among all workers on the broiler production chain in the targeted area, 165 participants from broiler keepers and 172 broiler meat workers (in total 337) were surveyed between 9 June 2021, and 2 September 2021. Prior to the interviews, we trained Arabic-speaking interviewers for two days about the study objectives, ethical considerations, and the Mobile Data Studio program for data collection. To reduce interviewer bias, we did not share the hypothesis that hygiene practices in broiler production might be inadequate with the interviewers. All data were collected electronically using the software program Mobile Data Studio and regularly transferred to a second electronic database and backup in Palestine and Switzerland.

### 4.4. Data Management and Analysis

All multi-stakeholder discussion groups from July 2021 were transcribed in Arabic and translated into English. The software program MAXQDA 2018 allowed for computer-assisted qualitative data analysis. Using a deductive approach, themes, categories, and codes were developed and assigned to the corresponding contents.

Data collected through surveys were managed using Stata version 16.1. (StataCorp. 2019. Stata Statistical Software: Release 16. College Station, TX: StataCorp LLC). The KAP of hygiene in the broiler production chain was assessed through descriptive analysis. Additionally, we examined the coherence between attitudes and practices using simple logistic regressions. For these simple logistic regressions, responses “No” and “Don’t know” to questions “Do you think avoiding contact with poultry can prevent infection with zoonotic diseases?” and “Do you think personal protective measures when in contact with poultry can prevent infection with zoonotic diseases?” were combined into one category. Further, using multiple logistic regression, education level and years of experience in farming were examined as determinants of respondent knowledge about zoonotic diseases. For this multiple logistic regression, the educational levels of “illiterate,” “elementary school”, and “junior high school” were grouped together and served as the reference category. Supplementary KAP of hygiene in broiler production were examined as risk factors for contracting chicken-borne zoonotic disease through simple logistic regressions.

## 5. Conclusions and Recommendations

Our study found that the overuse of antimicrobials, system fragmentation, insufficient infrastructure, a lack of regulations and controls, and poor hygiene practices are among the main obstacles to improving food safety in Palestine. To support the One Health Initiative in Palestine, we recommend promoting education, training, and awareness campaigns. One Health knowledge should be taught in multidisciplinary programs at universities. Furthermore, target groups such as farm workers, slaughterhouse and restaurant staff, households, and health workers should be educated and trained in sanitation practices and in the use of antimicrobials. Meanwhile, the general public’s awareness of food safety, zoonoses, and AMR needs to be raised through awareness campaigns.

Any development of the food safety system in Palestine should be accompanied by conducting research to demonstrate the added value of applying a One Health approach and to consider the growing challenges of climate change and the risks of epidemics. To effectively and sustainably monitor and control outbreaks of zoonotic diseases and the spread of AMR in Palestine, a national surveillance system must be established. Human, animal, and environmental samples need to be collected regularly and tested for zoonotic pathogens and their resistance rates, as well as for antimicrobial residues. Efficient testing relies on sufficiently sensitive detection methods to enable early detection, notification, and timely response.

As the next steps toward an operational One Health strategy for Palestine, we propose to further promote exchanges between stakeholders and establish an integrated quality control system for food production. These discussion groups aim to clarify the roles of all actors, establish communication pathways, and foster cross-sector partnerships. A One Health strategy for Palestine does not require new ministries or other institutions, as they are already in place and operational. Ideally, however, a governing body with representatives from the private and public sectors as well as academia should be established to oversee food safety in Palestine.

## Figures and Tables

**Figure 1 antibiotics-11-01359-f001:**
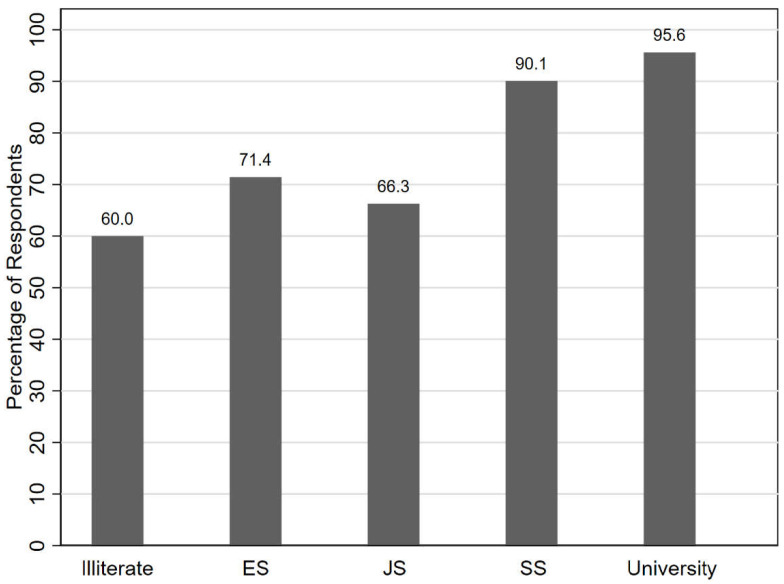
Percentage of respondents who have heard of zoonotic diseases by education level. Abbreviations: ES, Elementary school; JS, Junior high school; SS, Senior high school.

**Figure 2 antibiotics-11-01359-f002:**
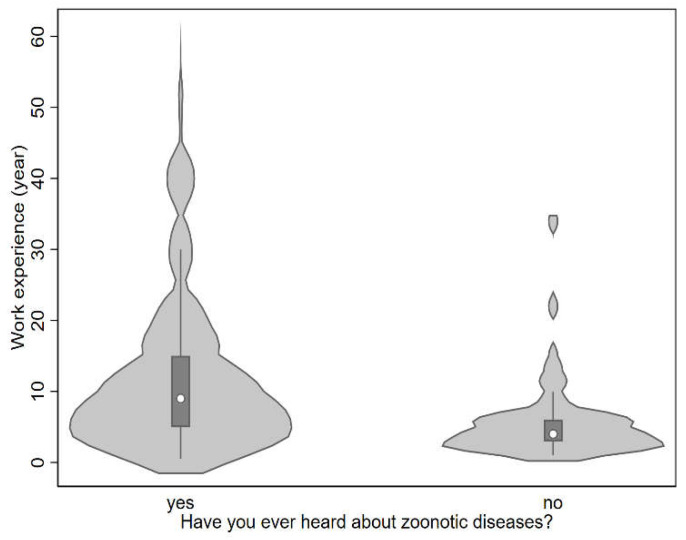
Years of experience working in the chicken meat production chain, by whether respondents have heard of zoonotic diseases or have never heard of them before.

**Figure 3 antibiotics-11-01359-f003:**
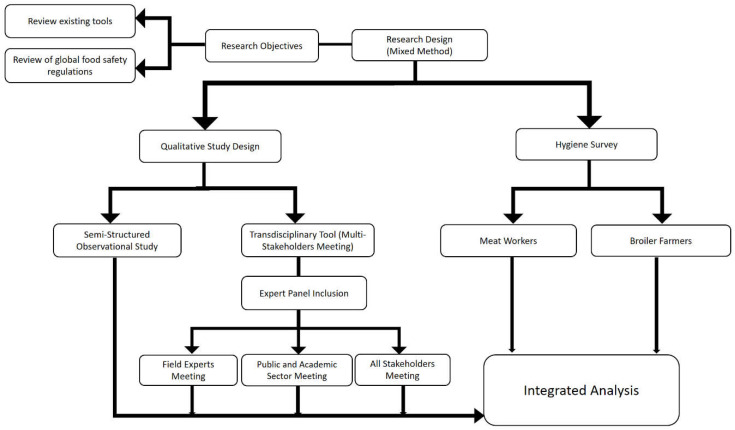
Methodology framework to assess the hygiene practices from farm to fork.

**Table 1 antibiotics-11-01359-t001:** Stratified the sociodemographic characteristics of the broiler workers surveyed based on workers’ profession/occupation.

	Farmers *N* (%)	Meat Workers *N* (%)	Overall *N* (%)
**Gender**			
Male	160 (97%)	171 (99.4%)	331 (98.2%)
Female	5 (3%)	1 (0.6%)	6 (1.78%)
**Age** (years)			
15–30	41 (24.9%)	85 (49.4%)	126 (37.4%)
31–45	58 (35.2%)	65 (37.8%)	123 (36.5%)
46–60	43 (26%)	15 (8.7%)	58 (17.2%)
>60	23 (19.9%)	7 (4.1)	30 (8.9%)
**Marital status** ^a^			
Single	30 (18.3%)	68 (39.5%)	98 (29.2%)
Married	133 (81.1%)	102 (59.3%)	235 (69.9%)
Divorced	1 (0.6%)	2 (1.2%)	3 (0.9%)
**Family Size** [median (IQR)]	6 (3)	5 (2)	5 (3)
**Residence**			
Urban	5 (3%)	29 (16.8%)	34 (10.1%)
Rural	159 (96.4%)	142 (82.6%)	301 (89.3%)
Refugee camp	1 (0.6%)	1 (0.6%)	2 (0.6%)
**Education level**			
Illiterate	3 (1.8%)	2 (1.2%)	5 (1.5%)
Elementary school (≤5 years of education)	7 (4.2%)	7 (4%)	14 (4.2%)
Junior high school (≤9 years of education)	37 (22.4%)	49 (28.5%)	86 (25.5%)
Senior high school (10–12 years of education)	62 (37.6%)	79 (45.9%)	141 (41.8%)
University (≥13 years of education)	56 (34%)	35 (20.4%)	91 (27%)
**Experience in farming** (years) [median (IQR)]	10 (15)	6 (7)	8 (11)
**Monthly income** (NIS) ^b^ [median (IQR)]	4500 (3000)	4000 (2200)	4000 (3000)

*N*, number of respondents; IQR, interquartile range. ^a^: One missing data in the marital status variable. ^b^: Sixteen missing data in the monthly income variable.

**Table 2 antibiotics-11-01359-t002:** Hygiene and zoonotic diseases-related knowledge and practices ^1^.

		Farmers *N* (%)	Meat Workers *N* (%)	Overall *N* (%)	*p*-Value for Difference ^1^
Do you think promoting proper hand hygiene can prevent infection with zoonotic diseases?	Yes	150 (91%)	166 (96.5%)	316 (93.8%)	*0.03*
	No	15 (9%)	6 (3.5%)	21 (6.2%)	
Do you think touching sick or dead poultry can cause infection with zoonotic diseases?	Yes	124 (75.2%)	140 (81.4%)	264 (78.3%)	*0.007*
	No	37 (22.4%)	20 (11.6%)	57 (16.9%)	
	Don’t know	4 (2.4%)	12 (7%)	16 (4.8%)	
Do you think increase the frequency of cleaning and disinfection and make sure there is adequate ventilation in shared spaces can prevent infection with zoonotic diseases?	Yes	158 (95.8%)	162 (94.2 %)	320 (95%)	*0.37*
	No	2 (1.2%)	6 (3.5%)	8 (2.4%)	
	Don’t know	5 (3%)	4 (2.3%)	9 (2.6%)	
Do you think personal protective measures when in contact with poultry can prevent infection with zoonotic diseases?	Yes	132 (80%)	146 (84.9%)	278 (82.5%)	*0.06*
	No	28 (17%)	16 (9.3%)	44 (13%)	
	Don’t know	5 (3%)	10 (5.8%)	15 (4.5%)	
How do you wash your hands?	Carefully, without soap	6 (3.6%)	32 (18.6%)	38 (11.3%)	*<0.0001*
	With soap or hand disinfectant	159 (96.4%)	140 (81.4%)	299 (88.7%)	
Did you touch sick or dead poultry with your hands in the last month?	Yes	105 (64%)	59 (34%)	164 (48.8%)	*<0.0001*
	No	59 (36%)	113 (66%)	172 (51.2%)	
Did you take preventive measures when you touched sick or dead poultry?	Yes	60 (58.8%)	28 (47.5%)	88 (54.7%)	*0.16*
	No	42 (41.2%)	31 (52.5%)	73 (45.3%)	
Have you ever heard about zoonotic diseases?	Yes	145 (88%)	139 (80.2%)	284 (84.3%)	*0.08*
	No	20 (12%)	33 (19.2%)	53 (15.7%)	
Have you experienced a disease problem regarding chicken health (zoonotic)?	Yes	11 (6.7%)	12 (7%)	23 (6.8%)	*0.91*
	No	154 (93.3%)	160 (93%)	314 (93.2%)	
How are the remains of slaughter and dead chickens handled?	Burned	18 (10.9%)	5 (2.9%)	23 (6.8%)	*<0.0001*
	Buried	11 (6.7%)	28 (16.3%)	39 (11.6%)	
	Fed to pets (dog or cat)	54 (32.7%)	23 (13.4%)	77 (22.9%)	
	Released into the wild	45 (27.3%)	32 (18.6%)	77 (22.9%)	
	Collected by the municipality	32 (19.4%)	77 (44.8%)	109 (32.3%)	
	Other	5 (3%)	7 (4%)	12 (3.5%)	

^1^ Chi2 test. *N*, number of respondents.

**Table 3 antibiotics-11-01359-t003:** The impact of SARS-CoV-2 on hygiene practices among farmers and meat workers in the chicken meat production chain.

What Protective Measures Have You Adopted in Your Work After SARS-CoV-2 Emerged?			
Protective Measures	Farmers *N* (%)	Meat Workers *N* (%)	Overall *N* (%)
Wearing masks	86 (52.1%)	133 (77.3%)	219 (65.0 %)
Wearing gloves	87 (52.7%)	143 (83.1%)	230 (68.2%)
Wearing overalls	74 (45.0%)	96 (55.8%)	170 (50.5%)
Frequent hand washing	149 (90.9%)	170 (98.8%)	319 (95.0%)
Sanitizing with disinfectant after finishing work	140 (84.9)	127(73.8%)	267 (79.2%)
Increasing the frequency of disinfection at the workplace	114 (69.1%)	162 (94.7%)	276 (82.1%)
Ventilation at the workplace	127 (77.0%)	160 (93.0%)	287 (85.2%)
Using a handkerchief when sneezing	108 (65.5%)	144 (83.7%)	252 (75.8%)

*N*, number of respondents.

**Table 4 antibiotics-11-01359-t004:** Number and row percentage of respondents (*N* (%)) stratified by attitude and related practice regarding hand hygiene and contact with poultry.

		How do you wash your hands?		
		Carefully, without soap	With soap or hand disinfectant	*Total*
Do you think promoting proper hand hygiene can prevent infection with zoonotic diseases?	Yes	36 (11.4%)	280 (88.6%)	*316 (100%)*
	No	2 (9.5%)	19 (90.5%)	*21 (100%)*
		Did you touch sick or dead poultry with your hands in the last month?		
		Yes	No	*Total*
Do you think touching sick or dead poultry can cause infection with zoonotic diseases?	Yes	134 (50, 8%)	130 (49.2%)	*264 (100%)*
	No	24 (42.9%)	32 (57.1%)	*56 (100%)*
	Don’t know	6 (37.5%)	10 (62.5%)	*16 (100%)*
		Did you take preventive measures when you touched sick or dead poultry?		
		Yes	No	*Total*
Do you think personal protective measures when in contact with poultry can prevent infection with zoonotic diseases?	Yes	74 (55.2%)	60 (44.8%)	*134 (100%)*
	No	8 (53.3%)	7 (46.7%)	*15 (100%)*
	Don’t know	6 (50%)	6 (50%)	*12 (100%)*

## Data Availability

The data presented in this study are available from the corresponding author upon reasonable request.

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
