# Peer review of "Towards a One Health Food Safety Strategy for Palestine: A Mixed-Method Study"

_antibiotics, 2022, doi:10.3390/antibiotics11101359_

Round 1

Reviewer 1 Report

It is interesting report on problems related to food safety in poultry production in Palestine involving the different stakeholders, however maybe a deeper description of the socio-economic conditions and  food chain system in Palestine should be added as you say:    "A One Health approach to foster food safety is a key for improvement, particularly in complex socio-ecological systems like in Palestine..."

I think that in the introduction a description of the  Palestinian social economic conditions should be better specified. 

Author Response

Response: We appreciate your comment and thank you for your feedback.

We included an explanation of the social and economic circumstances in Palestine in the introduction in response to your insightful feedback.

“Conflict zones, such as the ongoing conflict situation in Palestine, can have a negative impact on social structures, including the family and community. Such conflict scenarios can be extremely complicated and have an impact on aspects such as the environment, nutrition, economy, and psychosocial factors (23). The cost of war and violence results in environmental concerns in addition to psychological misery. Efforts to improve health care and public health will be hampered by this complicated socio-ecological system, which will also hinder social performance and economic progress (23, 24). In Palestine, 29.2% of the population, or more than one in three, were deemed to be living in poverty in 2017. In 2017, the poverty rate in the Gaza Strip was 53%, which was more than four times higher than the rate in the West Bank (25). By 2020, the agricultural sector's value added in Palestine has reached 7.1% of the gross domestic product (26).

Reviewer 2 Report

In the manuscript entitled “Towards a One Health Food Safety Strategy for Palestine: A Mixed-Method Study,” Abukhattab and colleagues present a mixed-method study “to identify the structure, strengths, obstacles, and limitations of the food production monitoring system in Palestine through multi-stakeholder discussion groups."

A study that describes the food production system in Palestine is very much worth to be published.

The review is informative and covers the aspects phrased in the aim. The results from the hygiene survey are interesting and show failures between public, private, and productive sectors and veterinarians. This is the first study that describes the knowledge, attitude, and prevention practices in productive systems in Palestine.

However, some conclusions are necessary to improve: “Our study has found that poor hygiene practices, the uncontrolled use of antimicrobials, system fragmentation, insufficient infrastructure, and a lack of regulations and controls are among the main obstacles to improved food safety in Palestine.” The statement “the uncontrolled use of antimicrobials” is not supported by recollected data, because in section 3.1.3. “Public Health,” authors reported opinions from stakeholders about AMR. Thus, the stakeholder's meeting showed an important problem on antimicrobial use in broiler chicken; still, it did not indicate an “uncontrolled use of antimicrobials.” Please state correctly.

Minor observations                        

1.     Correctly cite the references (avoid the use of capital letters and acronyms).

2.     Punctuation faults. Section 2.1.

3.     Remove extra spaces between words. Section 3.1.4.

4.     Remove extra spaces between words. Section 3.1.5.

5.     Correct the second row of table 3.

6.     Remove extra spaces between words. Discussion section.

7.     Organize the information provided in supplementary table 1 in a table format.

Author Response

Comment 1: The statement “the uncontrolled use of antimicrobials” is not supported by recollected data, because in section 3.1.3. “Public Health,” authors reported opinions from stakeholders about AMR. Thus, the stakeholder's meeting showed an important problem on antimicrobial use in broiler chicken; still, it did not indicate an “uncontrolled use of antimicrobials.” Please state correctly.

Response: It is a very important point, and thank you very much for this comment.

As you mentioned above in your comment in the public health section and the other discussion group outcomes, the majority of the participants emphasized the deficiencies in the mentor and control system and how using antibiotics has been impacted by these weaknesses.  However, your feedback is crucial; therefore, we have changed it to "overuse" instead of "uncontrolled" to prevent reader confusion.

Comment 2: Correctly cite the references (avoid the use of capital letters and acronyms).

Response: We attempted to correct it, and I assume you meant to reference number 58, but we were unable to do so because we use endnote software. So, this output of the endnote program.

Comment 3: Punctuation faults. Section 2.1.

 Response: Many thanks, Modified all of them.

Comment 4, 5, and 7: Remove extra spaces between words. Section 3.1.4., 3.1.5., and Discussion section

Response: Many thanks; we removed all of the extra spaces in the section 3.1.4., 3.1.5., and Discussion section.

Comment 6: Correct the second row of table 3.

Response: Thanks for this notice; we corrected the second row by adding the title “protective measures”.

Comment 7: Organize the information provided in supplementary table 1 in a table format

Response: Supplementary table 1 has been changed to table; many thanks once again.

Reviewer 3 Report

The paper survey the food safety from farm to public health toward an operational One Health strategy for Palestine using a transdisciplinary approach including multi-stakeholder discussion groups and a semi-structured observational tool. They find that misuse of antimicrobials, system fragmentation, insufficient infrastructure, a lack of regulations and controls, and poor hygiene practices are among the main obstacles to improving food safety in Palestine. And put forward solutions according to the above problems. The paper is quite elaborate. We recommend direct acceptance of the article.

Author Response

Comment 1: The paper survey the food safety from farm to public health toward an operational One Health strategy for Palestine using a transdisciplinary approach including multi-stakeholder discussion groups and a semi-structured observational tool. They find that misuse of antimicrobials, system fragmentation, insufficient infrastructure, a lack of regulations and controls, and poor hygiene practices are among the main obstacles to improving food safety in Palestine. And put forward solutions according to the above problems. The paper is quite elaborate. We recommend direct acceptance of the article.

Response: Thank you for your feedback and we appreciate your comment.